# Is there scope to improve the selection of patients with alcohol-related liver disease for referral to secondary care? A retrospective analysis of primary care referrals to a UK liver centre, incorporating simple blood tests

Freya Alison Rhodes ![ORCID],[1] Sara Cococcia,[2] Preya Patel ![ORCID],[1]
Jasmina Panovska-Griffiths ![ORCID],[3,4] Sudeep Tanwar,[1,5] Rachel H Westbrook,[1]
Alison Rodger,[6] William Rosenberg[1]

FAR and SC are joint first authors.

For numbered affiliations see end of article.

**Correspondence to**
William Rosenberg;
w.rosenberg@ucl.ac.uk

## ABSTRACT

**Objectives** Twenty per cent of people with alcohol use disorders develop advanced fibrosis and warrant referral to secondary care. Improving outcomes in alcohol-related liver disease (ArLD) relies on its earlier detection in primary care with non-invasive tests (NIT). We aimed to determine the proportion of alcohol-related referrals who were diagnosed with advanced fibrosis in secondary care, the prevalence of both alcohol and fatty liver disease ('BAFLD') and the potential impact of NIT on referral stratification.

**Design/setting** Retrospective analysis of all general practitioner-referrals with suspected ArLD/non-alcoholic fatty liver disease (NAFLD) to a UK hepatology-centre between January 2015 and January 2018.

**Participants** Of 2944 new referrals, 762 (mean age 55.5±13.53 years) met inclusion criteria: 531 NAFLD and 231 ArLD, of which 147 (64%) could be reclassified as 'BAFLD'.

**Primary outcome measure** Proportion of referrals with suspected ArLD/NAFLD with advanced fibrosis as assessed by tertiary centre hepatologists using combinations of FibroScan, imaging, examination and blood tests and liver histology, where indicated.

**Secondary outcome measures** Included impact of body mass index/alcohol consumption on the odds of a diagnosis of advanced fibrosis, and performance of NIT in predicting advanced fibrosis in planned post-hoc analysis of referrals.

**Results** Among ArLD referrals 147/229 (64.2%) had no evidence of advanced fibrosis and were judged 'unnecessary'. Advanced fibrosis was observed in men drinking ≥50 units per week (U/w) (OR 2.74, 95% CI 1.51 to 5, p=0.001) and ≥35 U/w in women (OR 5.11, 95% CI 1.31 to 20.03, p=0.019). Drinking >14 U/w doubled the likelihood of advanced fibrosis in overweight/obesity (OR 2.11; 95% CI 1.44 to 3.09; p<0.001). Use of fibrosis 4 score could halve unnecessary referrals (OR 0.50; 95% CI 0.32 to 0.79, p=0.003) with false-negative rate of 22%, but was rarely used.

## Strengths and limitations of this study

► This study reflects real-world experience of consecutive alcohol referrals from primary care to a specialist liver centre over a 3-year period.
► Results of tests routinely performed in primary care can be used to improve selection of patients for referral.
► This was a retrospective study relying on data held in electronic clinical records, including of self-reported alcohol intake.
► Our study used consensus judgement of expert hepatologists to assess liver disease rather than liver biopsy as a reference standard to assess fibrosis severity.

**Conclusions** The majority of referrals with suspected ArLD were deemed unnecessary. NIT could improve identification of liver damage in ArLD, BAFLD and NAFLD in primary care. Anecdotal thresholds for harmful drinking (35 U/w in women and 50 U/w in men) were validated. The impact of alcohol on NAFLD highlights the importance of multi-causality in chronic liver disease.

## INTRODUCTION

Approximately 90% of all chronic liver disease (CLD) is preventable, with the most common causes of cirrhosis attributed to alcohol-related liver disease (ArLD) and non-alcoholic fatty liver disease (NAFLD).[1] Mortality from cirrhosis has increased 400% since 1970, predominantly due to alcohol, although the rising prevalence of NAFLD is contributory.[2] Hepatic steatosis develops in up to 90% of people with alcohol use disorder (AUD) or obesity,[3 4] but advanced fibrosis or cirrhosis will affect only approximately 20%

of people with AUD[5] and 5% with NAFLD.[6] Both AUD and obesity can be managed effectively in primary care but advanced fibrosis and cirrhosis warrant management by liver specialists in secondary care. Detecting the minority of patients requiring specialist care is challenging because advanced fibrosis and most cases of cirrhosis are asymptomatic and simple liver function tests (LFTs) and ultrasound imaging are neither sensitive nor specific in detecting advanced fibrosis or cirrhosis.[7] As a consequence, three-quarters of people with CLD first present to healthcare with established advanced liver disease when behaviour change or therapeutic interventions have only modest impacts on prognosis.[1 8 9]

Conversely, as many as 92% of people referred to secondary care with suspected CLD do not have advanced fibrosis or cirrhosis requiring specialist care and could have remained in primary care for ongoing management.[10] Pathways of care employing the use of non-invasive tests (NITs) for liver fibrosis 4 score (FIB-4) and the enhanced liver fibrosis (ELF) test in primary care have been shown to be effective in the management of NAFLD, yielding an 88% reduction in 'unnecessary referrals' to liver specialists with a fivefold increase in the detection of advanced fibrosis and cirrhosis, and significant cost savings,[10 11] influencing national guidelines.[12] However, the proportion of referrals with AUD who do not have advanced ArLD that could be considered 'unnecessary' is unknown.

The ELF test has also been used successfully to triage patients from primary to secondary care with AUD in Denmark.[13] While current UK national guidelines recommend consideration of NIT in people with AUD in primary care,[7] alcohol pathways employing NIT are not widely established in the UK and none have been evaluated to our knowledge.

Although NAFLD and ArLD are described as distinct entities for research purposes, the risk factors for both conditions coexist in many patients. Moreover, it is increasingly recognised that alcohol and fat interact to cause liver damage, with obese people having increased risks of liver fibrosis for any given alcohol intake.[7 14–18] In this study we aimed to determine the proportion of patients referred for investigation of ArLD from primary care to secondary care hepatology clinics that had evidence of advanced fibrosis; and the prevalence of both alcohol and fat as co-contributing factors to CLD, termed both alcohol and fatty liver disease ('BAFLD') to describe the combination of BAFLD.[19] In addition, we aimed to determine the performance of simple NITs in the identification of cases of advanced fibrosis.

## METHODS

### Study design

This is a retrospective cross-sectional analysis of consecutive patients aged ≥18 years newly referred from primary care to a hospital-based hepatology service at the Royal Free London (RFL) NHS Foundation Trust, with a suspected diagnosis of ArLD or NAFLD between January 2015 and January 2018. Patients were excluded if they had any other hepatological diagnosis made prior to referral (online supplemental table 1).

### Outcome measures

The primary outcome measure was the proportion of new patients referred from general practitioner (GP) to hepatology clinic with suspected ArLD that had advanced fibrosis and could be deemed 'necessary' referrals.

Secondary outcome measures included the prevalence of 'BAFLD' among patients referred with suspected ArLD or NAFLD, analysis of demographic data as potential risk factors for a diagnosis of advanced fibrosis (including body mass index (BMI), alcohol consumption, smoking status, age, sex and deprivation score) and a post-hoc analysis of the performance of FIB4 and AST to Platelet Ratio Index (APRI) in predicting a diagnosis of advanced fibrosis.

### Study population

All electronic GP referrals for suspected ArLD or NAFLD during this period were reviewed in order to identify cases referred for NAFLD who were subsequently found to be drinking hazardous amounts of alcohol (>14 units per week (U/w)). As these conditions were not always reliably coded and triaged from the outset, every new referral from GP to hepatology clinic during this time period was reviewed in order to select out the NAFLD and ArLD referrals to ensure cases were not missed. Sample size was based on 3 years' worth of referrals.

'Suspected ArLD' referrals were defined as those in which the GP referral letter requested an assessment by a liver specialist specifying concerns about suspected ArLD or expressing concerns about a patient's alcohol intake.

'Suspected NAFLD' referrals were defined as those in which the GP referral letter either specified that they were referring the patient to hepatology 'with suspected NAFLD' or 'on the local NAFLD referral pathway', OR, in the absence of any other cause of liver dysfunction, where the GP specified that the patient had steatosis or CLD on ultrasound in combination with mentioning metabolic risk factors (BMI ≥25, diabetes, high waist circumference, high cholesterol or hypertension).

### Data collection

Anonymised data were extracted from the patients' electronic records. These included demographics, reason for referral, deprivation score, weight, height, waist circumference, alcohol intake, comorbidities and any fibrosis assessment before and after referral. Where weight and height were unavailable, but clinical records reported that the patient was overweight or obese, they were categorised accordingly to BMI >25 (overweight) or BMI >30 (obese). FIB4 and APRI scores were calculated using the blood tests from the first attendance to clinic after referral.

The diagnosis of advanced fibrosis (equivalent to a histological stage of ≥F3/4) or cirrhosis (≥F4) was established by expert clinical judgement by hepatologists based on a composite of FibroScan, imaging, blood tests, clinical examination and liver histology, where available, and this information was extracted from the electronic medical records. In the minority of cases where a diagnosis of advanced fibrosis was not clearly documented, decisions were reviewed by the study team (FAR and SC) and consensus achieved. FibroScan was considered diagnostic for advanced fibrosis if the elasticity of a valid scan was ≥11 kPa in ArLD[12 20] and ≥10 kPa in patients with NAFLD.[21] For variables where any data were missing, the denominator used in the analysis was adjusted for only available data.

'Unnecessary referrals' were defined as those patients that, subsequent to an assessment by a liver specialist, were deemed not to have advanced fibrosis and could be discharged back to ongoing care in the community.

In light of the frequent overlap between the two conditions, patients were subsequently recoded as having BAFLD if ArLD and NAFLD risk factors were both present. More specifically, BAFLD was applied to patients referred for suspected NAFLD who were subsequently found to be drinking more than 14 units of alcohol per week; and to patients who were referred for suspected ArLD, who also had either a BMI >25, or features of the metabolic syndrome. The metabolic syndrome was defined according to the International Diabetes Federation and American Heart Association as the presence of at least three of the following criteria: enlarged waist circumference (≥94 cm in European men, ≥90 cm in South Asian men and ≥80 cm in women), hypercholesterolaemia, hypertension and type 2 diabetes.[22]

## Statistical analysis

Descriptive statistical analyses included calculations of the frequencies and percentages for categorical variables, while for continuous data means and SD for normally distributed data, or medians and IQR for skewed data were used. For the comparison of categorical variables, $\chi^2$ or Fisher's exact test was used (the latter when n=<5), and for continuous data Mann-Whitney U test or Student's t-test depending on the data distribution.

For data with more than three variables to compare, analysis of variance (ANOVA) or Kruskall-Wallis ANOVA were used, depending on the distribution of the data.

Alcohol consumption was categorised into groups of U/w according to the perceived risk of liver damage established in the literature[7] (0–35, 36–50, 51–100, >100 U/w) and into quartiles of the population distribution of alcohol consumption for the ArLD cohort in which few patients were drinking <50 U/w. Multiple binary logistic regression analysis was used to determine the association between key variables and the presence of advanced fibrosis. The key variables were those risk factors for fibrosis that were of established importance in the literature, and those associated with p values <0.25 in the

univariate analysis. All p values were two-sided and significance set at <0.05. All data were analysed using SPSS software (V.25.0), except for the ORs for differences in outcomes for modelling of data with FIB4 compared with current practice, together with 95% CIs and $\chi^2$ for statistical significance which were performed using MedCalc statistical software 2018.

## Ethics

This study uses secondary anonymised patient data. The project was registered with the Integrated Research Application System (IRAS 272448) and judged to not require ethical approval or informed consent according to Health Research Authority guidance as it comprises data that were collected routinely as part of a registered service evaluation at the Royal Free London NHS Foundation Trust.

## Patient and public involvement

Patients and the public were not involved in this study.

# RESULTS
## Patient demographics

Between January 2015 and January 2018, a total of 2944 patients were referred to the RFL hepatology service from primary care and of these, 762 (mean age 55.5±13.53 years) met the inclusion criteria for this study; 231 patients were referred with suspected ArLD (mean age 54.68±12.37 years), and 531 with suspected NAFLD (mean age 55.88±14 years). One patient was deemed to have active hepatitis C virus infection as comorbidity and three were found to have inactive chronic hepatitis B after referral. The demographic characteristics of the included patients are reported in table 1. There was a higher proportion of male patients in the ArLD group (76.2%) than among the NAFLD group (54.2%, p<0.001). Active or previous smoking was significantly more common among those referred for ArLD compared with the NAFLD group (47.1% vs 11.3%; p<0.001). The average BMI was significantly higher in the NAFLD group than the ArLD group (31.9 and 27.9 kg/m² respectively, p<0.001), while median alcohol consumption was significantly higher in the ArLD group at 70 U/w (42–135), compared with 0 U/w (0–7) in the NAFLD group. The majority of the study population lay within the lowest four deciles of deprivation, and no significant difference in levels of deprivation was seen when ArLD and NAFLD referrals were compared (p=0.326).

## Reasons for referral from primary care

The presence of hepatic steatosis on an ultrasound scan and abnormal LFTs were the most common reasons for referral to hepatology clinic regardless of the aetiology. These were followed by elevated ELF and FIB4 in the NAFLD cohort (38.2% and 16.9% respectively). Only 38/231 (16.4%) of patients with suspected ArLD had an NIT in primary care prior to referral (25 ELF

**Table 1**  Baseline characteristics

| Patient characteristics | Overall (n=762) | Suspected ArLD referrals* (n=231) | Suspected NAFLD referrals† (n=531) | |
|---|---|---|---|---|
| Age (mean; SD) | 55.52±13.53 | 54.68±12.37 | 55.88±14 | p=0.262 |
| Male, n (%) | 464 (60.9%) | 176 (76.2%) | 288 (54.2%) | p<0.001 |
| BMI (mean; SD) | 30.85±6.23 | 27.9±5.46 (n=174) | 31.9±6.15 | p<0.001 p<0.001 |
| >25, n (%) | 608/732 (83.1) | 149/211 (70.6) | 459/521 (88.1) | p<0.001 |
| >30, n (%) | 350/675 (51.9) | 56/185 (30.3) | 294/490 (60) | |
| Alcohol intake U/w (median, IQR) | 5 (0–42.75) | 70 (42–134.8) | 0 (0–7) | p<0.001 |
| N= | 738 | 226 | 512 | |
| Years of harmful drinking | | | | p<0.001 |
| Median (IQR) | 0 (0–3) | 20 (6–30) | 0 (0–0) | |
| Total, n= | 598 | 143 | 455 | |
| Diabetes, n (%) | 235/760 (30.9) | 38/231 (16.5) | 197/529 (37.2) | p<0.001 |
| Hypertension, n (%) | 397/761 (52.2) | 113/231 (48.9) | 284/530 (53.6) | p=0.236 |
| Hypercholesterolaemia, n (%) | 352/759 (46.4) | 81/231 (35.1) | 271/528 (51.3) | p<0.001 |
| Smoking status | | | | p<0.001 |
| Non-smoker, n (%) | 369/681 (54.2) | 65/204 (31.9) | 304/477 (63.7) | |
| Smoker, n (%) | 150/681 (22) | 96/204 (47.1) | 54/477 (11.3) | |
| Ex-smoker, n (%) | 162/681 (23.8) | 43/204 (21.1) | 119/477 (24.9) | |
| ALT, median (IQR) | 45 (30–67) | 47 (30–68) | 45 (30–67) | p=0.360 |
| N= | 761 | 231 | 530 | |
| Deprivation score rank median (IQR) | 11 314 (6451–17 642) | 10 648 (6100–17 464) | 11 637 (6578–17 761) | p=0.326 |
| Deprivation score decile | | | | p=0.264 |
| 1 | 51 (6.7%) | 12 (5.2%) | 39 (7.3%) | |
| 2 | 146 (25.9%) | 53 (28.1%) | 93 (24.9%) | |
| 3 | 134 (43.4%) | 42 (46.3%) | 92 (42.2%) | |
| 4 | 107 (57.5%) | 30 (59.3%) | 77 (56.7%) | |
| 5 | 101 (70.7%) | 33 (73.6%) | 68 (69.5%) | |
| 6 | 82 (81.5%) | 26 (84.8%) | 56 (80%) | |
| 7 | 64 (89.9%) | 17 (92.2%) | 47 (88.9%) | |
| 8 | 44 (95.7%) | 8 (95.7%) | 36 (95.7%) | |
| 9 | 22 (98.6%) | 6 (98.3%) | 16 (98.7%) | |
| 10 | 11 (100%) | 4 (100%) | 7 (100%) | |
| Had biopsy, n (%) | 122/762 (16) | 10/231 (4.3%) | 112/531 (21.1.%) | p<0.001 |
| Had FibroScan, n (%) | 575/762 (75.5%) | 158/231 (68.4%) | 417/531 (78.5%) | p=0.003 |
| Valid FibroScan reading‡ | 524/575 (91%) | 140/158 (89%) | 389/417 (93%) | |
| FibroScan median KPa (IQR) | 5.5 (4.5–7.7) | 6 (4.7–8.5) | 5.4 (4.4–7.5) | p=0.03 |

*Where primary reason for referral from GP was for suspected alcohol-related liver disease.
†Where primary reason for referral from GP was for suspected NAFLD.
‡FibroScan results were considered invalid if: IQR/M>30%, success rate <60%, <10 valid readings or if this information was not recorded in the FibroScan report (missing information about IQR/M ratio/success rate made up n=22/575 FibroScan results).
ALT, alanine aminotransferase; ArLD, alcohol-related liver disease ; BMI, body mass index; GP, general practitioner; NAFLD, non-alcoholic fatty liver disease ; U/w, units per week.

scores and 13 FIB4) and of these, 25/38 (66%) patients had comorbid features of the metabolic syndrome and so were subsequently recoded as BAFLD. Among the NAFLD referrals 293/531 (55.2%) had an NIT prior to referral in accordance with the local NAFLD pathway. Of these patients 203/293 (69%) were referred on the basis of an elevated ELF test and 90/293 (31%) based on their FIB4 score.

### Prevalence of advanced fibrosis in patients referred with suspected ArLD or NAFLD

Data on fibrosis stage were available for 758/762 patients following hepatology review, with 4 not attending for assessment. Of patients with suspected ArLD, 64.2% (147/229) had no evidence of advanced fibrosis and could be discharged back to primary care. This figure was even higher in the NAFLD cohort with 83.4% not having advanced fibrosis.

Of the patients referred with suspected ArLD who had advanced fibrosis (82/229), the frequency with which fibrosis tests were used were: liver biopsy in 10% (8/82), FibroScan in 41% (34/82) and radiology in 62% (51/82).

Of the patients referred with suspected NAFLD who had advanced fibrosis (88/529), the frequency with which fibrosis tests were used were: liver biopsy in 47% (41/88), FibroScan in 64% (56/88) and radiology in 33% (29/88).

### Risk of advanced fibrosis (>/F3) in patients referred with suspected ArLD

Univariate analysis of the 231 patients referred with ArLD revealed that advanced fibrosis was associated with raised alkaline phosphatase (ALP) (OR 1.012, 95% CI 1.006 to 1.018, p<0.001) and higher alcohol consumption (alcohol data available for 224/231) (OR 1.006, 95% CI 1.002 to 1.010, p=0.006). When categorised into alcohol unit groups of: <35 U/w, 36–50 U/w, 51–100 U/w, >101 U/w; patients drinking >50 U/w had a higher risk of advanced fibrosis in this cohort (OR 2.899, 95% CI 1.068 to 7.869, p=0.037). The multivariable logistic regression model found that the odds of advanced fibrosis in suspected ArLD was independently associated with increased units of alcohol consumed, (OR 1.007, 95% CI 1.002 to 1.012, p=0.007), ALP (OR 1.009, 95% CI 1.002 to 1.016, p=0.01)

and reduced platelets (OR 0.992, 95% CI 0.988 to 0.996, p<0.001). There was a trend towards higher odds of advanced fibrosis with increased age, but this did not reach significance (p=0.059).

### Patients with risk factors for both ArLD and NAFLD: 'BAFLD'

Patients with risk factors for both ArLD and NAFLD were classified as BAFLD (as defined earlier) and the whole cohort was reclassified into three categories: ArLD, NAFLD and BAFLD, in order to evaluate further risk factors for advanced fibrosis (figure 1).

From the GP referral letters, 147 (63.6%) patients out of the 231 patients referred to the hepatology clinic with suspected ArLD were overweight, or met the diagnostic criteria of the metabolic syndrome and were therefore reclassified as BAFLD. Of the 531 patients referred to hepatology as suspected NAFLD, 80 of them (15.1%) also regularly consumed an average of more than 14 U/w and were reclassified as BAFLD. Overall, 83.1% of the whole cohort were overweight and 50% obese. As expected, the proportion of patients who were overweight and obese was significantly higher in the NAFLD cohort compared with ArLD cohort (p<0.001). The main characteristics of the three cohorts can be found in online supplemental table 2.

Patients with BAFLD had almost double the prevalence of advanced fibrosis when compared with NAFLD (29% and 16.2%, respectively, (OR 2.11, 95% CI 1.441 to 3.094), p<0.001), suggesting that hazardous drinking doubled the risk of fibrosis in people who are overweight or obese in this study population.

Patients in the ArLD cohort had the highest prevalence of advanced fibrosis (38%), and their weekly alcohol intake was almost double that of the patients with BAFLD,

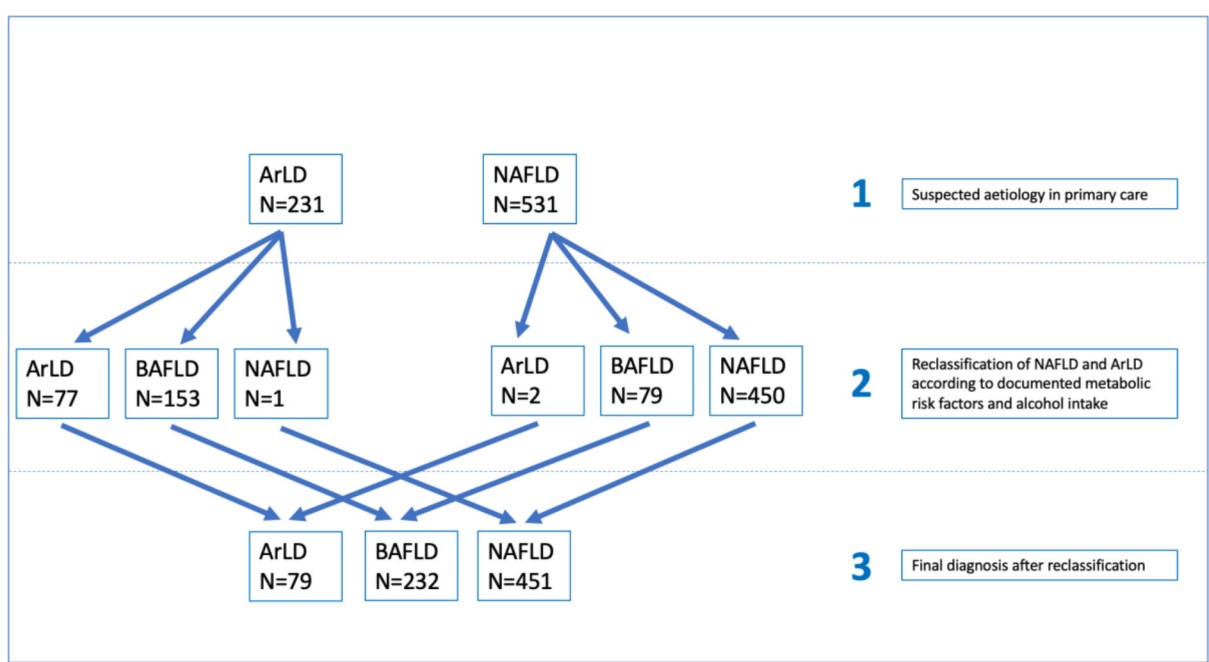

**Figure 1** Flow chart depicting reclassification of aetiologies. ArLD, alcohol-related liver disease; BAFLD, both alcohol and fatty liver disease; NAFLD, non-alcoholic fatty liver disease.

precluding the opportunity to compare the impact of overweight/obesity on heavy alcohol consumption in this cohort.

### Influence of alcohol on fibrosis risk

As the number of ArLD patients drinking <50 U/w was small, the entire cohort (n=762) was examined in an attempt to identify a potential threshold for the effect of alcohol on fibrosis risk. Other factors influencing fibrosis risk including age and BMI were also studied. Alcohol data were available for 734/762 patients.

Increased alcohol U/w predicted advanced fibrosis (OR 1.009, 95% CI 1.006 to 1.012, p =<0.001) on univariate analysis.

Alcohol units were categorised into quartiles of the reported distribution of consumption (0–42 U/w, 43–70 U/w, 71–135 U/w, >136 U/w). Binary logistic regression revealed that patients consuming ≥43 U/w were at greater risk of advanced fibrosis than those drinking less than 43 U/w (OR 1.814, 95% CI 1.038 to 3.172, p=0.037), and those drinking ≥70 U/w were at more than four times the risk of having advanced fibrosis compared with those drinking less than 43 U/w (OR 4.25, 95% CI 2.334 to 7.740, p =<0.001).

Alcohol consumption was then evaluated at literature-based unit thresholds of interest (0–35 U/w, 36–50 U/w, 51–100 U/w, >101 U/w) revealing that drinking more than 35 U/w was associated with double the odds of developing advanced fibrosis compared with those drinking <35 U/w (OR 2.173, 95% CI 1.119 to 4.219, p=0.022) and the odds increased to over fivefold in those drinking more than 100 U/w (OR 5.044, 95% CI 3.071 to 8.284, p<0.001).

A different threshold effect was found when these data were analysed separately for men and women. In the overall cohort of 762 patients, the risk of having advanced fibrosis was higher in those men drinking >50 U/w (OR 2.743, 95% CI 1.506 to 4.998, p=0.001), while in women the risk of having advanced fibrosis increased significantly at only >35 U/w (OR 5.115, 95% CI 1.306 to 20.030, p=0.019), compared with <35 U/w.

In the overall cohort of 762 patients with ArLD/NAFLD/BAFLD (of which complete data for this model were available for 625/762), multivariable regression analysis revealed that increased units of alcohol, age, ALP, BMI and decreased platelet count were significantly associated with increased odds of a diagnosis of advanced fibrosis.

### Modelling the impact of indirect fibrosis tests on the detection of advanced fibrosis in patients referred from primary care with suspected ArLD

Blood test results from the first attendance at the secondary care were used to calculate FIB4 and APRI scores for 225/231 patients referred with suspected ArLD (six patients did not have an aspartate aminotransferase (AST) value available). Median FIB4 and APRI were 1.58 (IQR 0.97–3.29) and 0.68 (IQR 0.36–1.53) respectively.

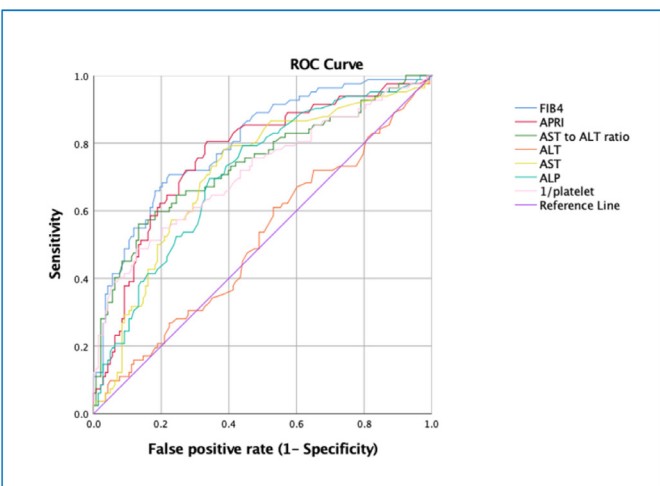

**Figure 2** ROC analysis of the performance of indirect tests for fibrosis and simple liver blood tests in the detection of advanced fibrosis (composite clinical judgement) in patients referred with suspected alcohol-related liver disease (N=231). AUROCs with 95% CI in brackets: FIB4: 0.801 (0.742 to 0.860); APRI: 0.763 (0.697 to 0.829); AST:ALT ratio: 0.739 (0.668 to 0.809); ALT: 0.512 (0.433 to 0.591); AST: 0.711 (0.640 to 0.782); ALP: 0.708 (0.638 to 0.777); 1/platelet: 0.714 (0.641 to 0.787). (All p values <0.001 apart from ALT which was non-significant at p=0.758). ALP, alkaline phosphatase; ALT, alanine aminotransferase; APRI, AST to Platelet Ratio Index; AST, aspartate aminotransferase; FIB4, fibrosis 4 score.

Both scores independently predicted the clinical diagnosis of advanced fibrosis in secondary care in multivariable regression analysis (for FIB4, OR=1.658, 95% CI 1.397 to 1.967, p<0.001; for APRI, OR=1.485, 95% CI 1.204 to 1.832, p<0.001).

When Receiver Operator Characteristic (ROC) curve analysis was used to examine the ability of NIT based on routine blood tests to predict a diagnosis of advanced fibrosis, FIB4 performed the best Area Under the Receiver Operator Characteristic Curve (AUROC 0.801), compared with APRI, AST, alanine aminotransferase (ALT), ALP and platelet count (all p<0.005 using DeLong comparison) and numerically but not significantly better than APRI (p=0.06) (figure 2).

Among the cohort of patients with ArLD referred to secondary care, 35.81% were judged to have advanced fibrosis and thus 64.2% could be considered 'unnecessary' referrals. Use of an FIB4 threshold of ≥3.25[23] could have improved the detection of patients with advanced fibrosis nearly fivefold (OR=4.82; 95% CI 2.56 to 9.09, p<0.0001), leading to a 79.3% reduction in unnecessary referrals to secondary care (64.2% to 27.1%) (OR=0.21; 95% CI 0.11 to 0.39, p<0.001) However, this would be associated with the exclusion of 39 patients judged to have advanced fibrosis (false negative rate of 47.6%) (table 2).

When modelling the referrals using an FIB4 threshold of ≥1.45,[23] the detection of advanced fibrosis improved twofold compared with standard care (OR=1.98; 95% CI 1.27 to 3.09, p=0.0027) and reduced the number of

**Table 2** Accuracy of indirect fibrosis markers in detecting advanced fibrosis in a cohort of 231 patients referred from primary care with suspected alcohol-related liver disease (N=225/231)

| Indirect fibrosis test (N=225/231) | Correctly classifies | Sensitivity (95% CI) | Specificity (95% CI) | PPV (95% CI) | NPV (95% CI) | LR+ (95% CI) | LR- (95% CI) | TP FP | FN TN | False negative rate (%) | False positive rate (%) |
|---|---|---|---|---|---|---|---|---|---|---|---|
| APRI ≥1 | 165 (73.3%) | 64.6% (54% to 75%) | 78.3% (70% to 85%) | 63.1% (52% to 73%) | 80% (72% to 86%) | 3.02 (2.13 to 4.28) | 0.44 (0.33 to 0.6) | 53 31 | 29 112 | 35.4 | 21.7 |
| FIB4 ≥3.25 | 170 (75.6%) | 52.4% (41.2% to 63.5%) | 88.8% (82.2% to 93.3%) | 72.9% (59.5% to 83.3%) | 76.5% (69.1% to 82.6%) | 4.69 (2.83 to 7.77) | 0.54 (0.43 to 0.67) | 43 16 | 39 127 | 47.6 | 11.2 |
| FIB4 ≥1.45 | 149 (66%) | 78% (67.3% to 86.1%) | 59.4% (50.9% to 67.4%) | 52.4% (43.3% to 61.5%) | 82.5% (73.5% to 89%) | 1.92 (1.53 to 2.42) | 0.37 (0.24 to 0.56) | 64 58 | 18 85 | 22 | 40.6 |

APRI, AST to Platelet Ratio Index; FIB4, fibrosis 4 score; LR, likelihood ratio; NPV, negative predictive value; PPV, positive predictive value.

unnecessary referrals from 64.2% to 47.5% (OR=0.5; 95% CI 0.32 to 0.79, p=0.003), with 103 patients (45.7%) having an FIB4 score below 1.45 that could have remained in primary care. The false negative rate was lower using FIB4 ≥1.45 compared with threshold ≥3.25 (18/103, 22% compared with 39/103, 47.5%; $X^2$=10.60; p=0.001).

## DISCUSSION

Two-thirds of the patients referred to secondary care for suspected ArLD had no evidence of advanced fibrosis, representing unnecessary referrals. This can be explained in part because the most common reasons for referral were abnormal LFTs and ultrasound scans, neither of which are sensitive or specific tests for advanced fibrosis.[7] While some of these patients may have benefited from a hepatologist's advice about the wider consequences of their drinking, many primary care physicians consider that they are better placed to deliver brief advice about hazardous or harmful drinking and referral to liver specialists should be restricted to patients with ArLD. Only 38/231 patients with suspected ArLD had any kind of fibrosis assessment prior to referral to secondary care, the majority of whom had features of metabolic syndrome or were overweight and received FIB4 and ELF tests suggesting that their GPs had followed the local NAFLD pathway that incorporates these investigations. These patients were reclassified as having BAFLD.

The majority (64%) of patients referred with suspected ArLD were overweight, obese or had features of metabolic syndrome. These patients with BAFLD had double the odds of advanced fibrosis when compared with the NAFLD cohort suggesting that hazardous drinking is associated with a doubling of the risk of liver fibrosis in people who are overweight or obese. This both highlights the increased risk of liver disease in patients with dual pathology and the importance of considering multimorbidity in CLD.

Although national guidelines state that the risk of advanced fibrosis develops at a lower alcohol unit threshold for women than men (<35 U/w for women and <50 U/w for men),[7] these thresholds are not based on published data that we have been able to identify. Few studies have investigated the association between levels of alcohol consumption and the risk of advanced fibrosis, and those that did have reported a range of thresholds.[24–29] Furthermore, the levels of drinking that cause harm in the context of overweight and obesity are not known but we derived these same thresholds of 35 U/w in women and 50 U/w in this cohort of 762 patients that included a high prevalence of overweight and obese people. It should be noted that these thresholds focus purely on the risk of advanced liver fibrosis and cannot be generalised to other health measures. National guidelines state that there is an increased risk to health above 14 U/w.

The performance of 'indirect' serum fibrosis tests is well reported in NAFLD, but less so in ArLD. In this

study cohort of 231 patients with ArLD, FIB4 and APRI outperformed simple liver blood tests (ALP, ALT, AST and platelet count) in predicting a diagnosis of advanced fibrosis on AUROC analysis, with FIB4 having the highest AUROC of 0.801. However, when examining FIB4 at literature-derived binary thresholds of 3.25 and 1.45,[13 23] it did not perform as well in detecting clinically defined advanced fibrosis as has been reported in a recent study in which all participants were required to undergo liver biopsy.[13] Stratifying patients in primary care using an FIB4 threshold of 3.25 could have reduced unnecessary referrals by 79.3%, with positive predictive value (PPV) and negative predictive value for the detection of advanced fibrosis of 72.9% and 76.5%, respectively. However, the associated false negative rate was 47.5% suggesting that nearly half the cases of advanced fibrosis would be left in primary care, making it unsuitable for case stratification. An FIB4 threshold of 1.45 produced a lesser, but still significant, false negative rate of 22%, and although it reduced the proportion of unnecessary referrals by 50%, the PPV was 52.4% and overall, this threshold correctly classified only 66% of patients into presence or absence of advanced fibrosis. These results suggest that an effective ArLD pathway would require the use of either an NIT with better diagnostic performance or the use of two or more NIT in series, as employed in the Camden and Islington NAFLD pathway.[10]

This retrospective study lacked access to liver biopsy as a reference standard to stage fibrosis severity. Self-reported alcohol intake at the point of referral to secondary care was used to record drinking behaviour and this may not be reliable. However, this clinic-based sample of 'real-world' cases reflects current practice in the UK and many other countries and highlights the opportunity to stratify patients with ArLD community settings to ensure that only those with a high likelihood of advanced fibrosis are referred for liver specialist care.

Having so many 'unnecessary referrals' to secondary care is not only an inefficient use of resources, but also exposes patients to unnecessary investigation and the associated time, risk and anxiety. These patients could be managed more appropriately in community settings with an appropriate focus on the wider harms associated with their drinking. Conversely emphasis on those with advanced fibrosis might improve the early detection of those drinkers who are likely to progress to cirrhosis and suffer life-limiting effects of their drinking.

Based on the performance of APRI and FIB4 in this cohort, we would not recommend their routine use to risk stratify patients with AUD. Instead, further evaluation of pathways incorporating non-invasive tests such as ELF or FibroScan[7 12 13] would be preferable.

This study highlights the multicausality and multimorbidity endured by patients with ArLD and NAFLD. Although the interaction between alcohol and obesity is recognised, the low threshold of alcohol consumption at which the risk of advanced fibrosis nearly doubled in this cohort highlights the importance of communicating this risk to patients with fatty liver disease in clinics and through public health messaging. There is a need for greater awareness among healthcare professionals, policy-makers and the public and a need for a multidisciplinary approach to address the lifestyle risk factors that are likely to influence the morbidity and mortality of those with BAFLD.

In summary, the current referral strategy for patients with AUDs at risk of liver disease from primary care is inefficient and ineffective. There is a need for increased awareness of the need to search for fibrosis using appropriate strategies incorporating non-invasive testing, and education of the guidelines for fibrosis testing in both AUD and NAFLD. In addition, there is a need for improved collaboration between primary and secondary care services to develop referral pathways employing NIT, with evaluation to further refine thresholds for referral and education to improve awareness and the advice provided to patient about the impact of overweight/obesity and alcohol on liver health.

**Author affiliations**
[1]Institute for Liver and Digestive Health, Division of Medicine, UCL, London, UK
[2]First Department of Internal Medicine, University of Pavia, Pavia, Italy
[3]Department of Applied Health Research, UCL, London, UK
[4]Institute for Global Health, UCL, London, UK
[5]Department of Gastroenterology, Barts Health NHS Trust, London, UK
[6]Department of Infection and Population Health, UCL, London, UK

**Contributors** FAR performed the analysis of the data, contributed to the collection of the data and wrote the first draft of the manuscript. SC contributed to the collection of data and to the writing of the manuscript. PP contributed to the review and editing of the manuscript. JP-G contributed to the statistical analysis. RHW, ST and AR contributed to the analysis methodology, and the review and editing of the manuscript. WR conceived the study, oversaw the study and reviewed and edited the manuscript drafts. All authors approved the final version of the paper. The corresponding author attests that all listed authors meet authorship criteria and that no others meeting the criteria have been omitted.

**Funding** This study is being supported by funding from WR's National Institute for Health Research (NIHR) Senior Investigator Award (award number 200249). WR is an NIHR Senior Investigator and is supported by the NIHR University College London Hospitals Biomedical Research Centre. JP-G was supported by the UK NIHR Applied Research Collaboration North Thames (ARC North Thames) at Bart's Health NHS Trust.

**Competing interests** WR is an inventor of the enhanced liver fibrosis test but receives no related royalties. WR has received speakers' fees from Siemens Healthineers.

**Patient consent for publication** Not required.

**Provenance and peer review** Not commissioned; externally peer reviewed.

**Data availability statement** Data are available upon reasonable request. On publication of this article, the data set will be made available from the corresponding author on reasonable request.

**ORCID iDs**
Freya Alison Rhodes http://orcid.org/0000-0002-6884-6235
Preya Patel http://orcid.org/0000-0002-2433-6794
Jasmina Panovska-Griffiths http://orcid.org/0000-0002-7720-1121

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
