## [Reviewer comments · BMJ Open]

ARTICLE DETAILS

TITLE (PROVISIONAL)	Is there scope to improve the selection of patients with alcohol-related liver disease for referral to secondary care? A retrospective analysis of primary care referrals to a UK liver centre, incorporating simple blood tests.
AUTHORS	Rhodes, Freya; Cococcia, Sara; Patel, Preya; Panovska-Griffiths, Jasmina; Tanwar, Sudeep; Westbrook, R; Rodger, Alison; Rosenberg, William

VERSION 1 – REVIEW

REVIEWER	Krag, Aleksander Department of Gastroenterology, Odense University Hospital, University of Southern Denmark
REVIEW RETURNED	04-Jan-2021

GENERAL COMMENTS	This is a retrospective analysis of referrals, 762 referrals from primary care to secondary care were analyzed, the study aimed to determine the proportion of patients referred for investigation of ArLD from primary-care to secondary-care hepatology clinics that had evidence of advanced fibrosis. A High proportion (64%) was deemed unnecessary, NIT was rarely used but could reduce unnecessary referral rate. This is a very relevant study that question the quality of current healthcare pathways and tools used to qualify need for referral to hepatology units among metabolic liver diseases. However, there are a number of issues that needs clarification. page 6/25: overall introduction: the structure of the introduction could be optimized- be shorter and more chronological - page 8/25: Study design could be clearer defined-page 8/45 Study population: how was ArLD or NAFLD identified among all referrals?- electronic search, manual referrals and what criteria should a referral meet to be included?- more precise on in/ex criteria.- page 8/25, line 40: how was the "suspected diagnosis" of liver disease defined?- page 8/25, line 43: how about other diagnoses? If they had ArLD and eg. cancer diagnosed?- page 8/25, line 47: How is it assessed that patients are referred for NAFLD? Does the GP states that he suspects NAFLD prior to referral?- page 8/25, line 53: How is the secondary outcomes
---

	defined? What are included in referral reasons and risk factors?  - page 9/25, line 14: how was missing values handled? - page 9/25, line 20: Did the authors use a minimum of required tests to establish consensus for the advanced fibrosis diagnosis? Otherwise, from the current text, the diagnosis could be established based solely on blood tests. Although it is stated only in a minority of cases. Please provide info on the basis for classifying into fibrosis category: how many based on biopsy, TE, other? -Choice of TE cut-off: always some controversy around this, however very important in particular for cross study comparison. The recent multi etiology large scale validation study (PMID: 33307138) provides some clarity and support 12Kpa as a general cut-off to rule-in advanced cirrhosis and 7kpa to rule out. With the current development in the NASH field and health care pathways some places towards treatment of lifestyle factors, significant fibrosis ($\geq F2$) is also relevant and should be included in the main analysis – i.e. cut-off of 8kpa. Further it must be described what defines an unnecessary referral - $<10kpa$?  - page 14/25, line 17: are the limits for alcohol consumption equal for men and women, with 14 units per week? - page 15/25, line 37: Why was blood tests from secondary care used? Was it not available in primary care at the referral point? Table 1: provide Number with liver biopsy, number with successful fibroscan and value of FS Table 2, also provide data for Fib4 of $> 2,67$
--	---

REVIEWER	Schattenberg, Joern Johannes Gutenberg University Hospital Mainz
REVIEW RETURNED	21-Jan-2021

GENERAL COMMENTS	This is an important analysis. It needs to be clarified, how alcohol consumption was captured and analyzed in this retrospective chart review. How was the classification for cases with undocumented alcohol consumption or uncertainty about alcohol consumption handled? Which categories did you use to classify in the absence of the documentation of gram / day in the medical records? The term BALFD is not well defined. Has it been adopted by an organization? Is there evidence, that the outcome is different when comparing to NAFLD or ALD? Please provide some rationale for addressing this. How is counselling different - with regards to prevention and if not, why single this out vs for example stratifying by the amount of fibrosis, e.g. non-significant fibrosis vs significant fibrosis? Here treatment and outcomes would be different. For discussion. This reviewer struggles with the term unnecessary referral, as it neglects the need and function to provide counseling in the absence of advanced fibrosis and in patients that the primary needs some "additional" feedback on. The term
--

	"misdirected" would catch this better? Meaning another specialist would be needed? If Hepatology declines this referrals, it should be clear where they should be sent to inform referring physicians. While the context (NICE system) is important, a management proposition would be of interest to non-NICE based physicians. Which specialist should deal with these patients? When should they be refereed to Hepatology? The reduction on prioriasation of referrals falls short to address the management need that arises. Please put this into perspective. Can you provide the reason for referral? Along this line, did you see all refereed patients or is there another filter that would decide to reject a referral based on a decision before reaching the clinic? How many referrals were rejected? Few typos need to be corrected.
--	---

VERSION 1 – AUTHOR RESPONSE

Reviewer: 1

Dr. Aleksander Krag, Odense University Hospital

Comments to the Author:

This is a retrospective analysis of referrals, 762 referrals from primary care to secondary care were analyzed, the study aimed to determine the proportion of patients referred for investigation of ArLD from primary-care to secondary-care hepatology clinics that had evidence of advanced fibrosis. A High proportion (64%) was deemed unnecessary, NIT was rarely used but could reduce unnecessary referral rate. This is a very relevant study that question the quality of current healthcare pathways and tools used to qualify need for referral to hepatology units among metabolic liver diseases.

However, there are a number of issues that needs clarification.

- 1) *Page 6/25: overall introduction: the structure of the introduction could be optimized- be shorter and more chronological*

We have tracked the changes to make the introduction more concise.

- 2) *page 8/25: Study design could be clearer defined*

We have clarified the definitions used of ArLD and NAFLD in the study, and the methods used to assign these diagnoses (page 7 lines 29-33, page 8 lines 1-4). Furthermore, the primary and secondary outcome measures are now clearly specified with new text added on page 7 lines 14-21. We hope that by addressing these points we have adequately explained the study design.

- 3) *page 8/45 Study population: how was ArLD or NAFLD identified among all referrals?- electronic search, manual referrals and what criteria should a referral meet to be included?- more precise on in/ex criteria.*

Electronic patient records were used to identify which were the ArLD and which were NAFLD GP referrals. As these clinical conditions were not always reliably coded in the referral letter, every new referral from GP to any hepatology clinic (for any reason) during the study time period was reviewed in order to identify the primary reason for referral as NAFLD or alcohol and to ensure that eligible cases were not missed.

'Suspected ArLD' referrals were defined as those in which the GP referral letter requested an assessment by a liver specialist, specifying concerns about suspected ArLD, or expressing concerns about a patient's alcohol intake.

'Suspected NAFLD' referrals were defined as those in which the GP referral letter either specified that they were referring the patient to a liver specialist 'with suspected NAFLD' or 'on the local NAFLD referral pathway', or, in the absence of any other cause of liver dysfunction, where the GP specified that the patient had steatosis or signs of chronic liver disease on ultrasound in combination with mentioning metabolic risk factors (BMI \geq 25, diabetes, high waist circumference, high cholesterol or hypertension).

We have now clarified the description of our use of electronic records, and included the two paragraphs above in the description of the 'study population' in the Methods section on pages 7 (lines 29-33) and page 8 (lines 1-4).

4) page 8/25, line 40: how was the "suspected diagnosis" of liver disease defined?

As clarified above in point 3, we interpreted the GPs' decision to refer the patients to a specialist liver clinic for review by a hepatologist as evidence that they had concerns about the patient's liver, and have now better described the definitions of suspected ArLD and NAFLD diagnoses (page 7 lines 29-33, page 8 lines 1-4).

5) page 8/25, line 43: how about other diagnoses? If they had ArLD and eg. cancer diagnosed?

Patients were excluded if they had a pre-existing liver diagnosis of any aetiology, including liver cancer, prior to referral to the hepatology clinic. Subsequent to referral, one patient was found to have co-existing hepatitis C, and three were deemed to have inactive chronic hepatitis B. This is documented in the manuscript under the 'patient demographics' results section, page 10, lines 10-13.

6) page 8/25, line 47: How is it assessed that patients are referred for NAFLD? Does the GP states that he suspects NAFLD prior to referral?

This is defined as explained in point 3) above. Locally, GPs can either specify suspected NAFLD in their referral, or use a designated NAFLD primary care pathway to refer the patient to clinic.

7) page 8/25, line 53: How is the secondary outcomes defined? What are included in referral reasons and risk factors?

We have now added a section entitled 'outcome measures' under the Methods section on page 7 (lines 14-21) which addresses this. It includes the following two paragraphs:

'The primary outcome measure was the proportion of new patients referred from GP to hepatology clinic with suspected ArLD that had advanced fibrosis and could be deemed 'necessary' referrals.

Secondary outcome measures included the prevalence of 'BAFLD' cases amongst patients referred with suspected ArLD or NAFLD, analysis of demographic data for potential risk factors for advanced fibrosis (including BMI, alcohol consumption, smoking status, age, sex, and deprivation score), and a post-hoc analysis of the performance of FIB4 and APRI in predicting a diagnosis of advanced fibrosis.'

8) page 9/25, line 14: how was missing values handled?

In response to this we have now added the following sentence to the methods section referenced to in this question, page 8 lines 21-22:

'For variables where any data were missing, the denominator used in the analysis was adjusted for only available data.'

9) page 9/25, line 20: *Did the authors use a minimum of required tests to establish consensus for the advanced fibrosis diagnosis? Otherwise, from the current text, the diagnosis could be established based solely on blood tests. Although it is stated only in a minority of cases. Please provide info on the basis for classifying into fibrosis category: how many based on biopsy, TE, other?*

The diagnosis of advanced fibrosis was based on the composite judgement of expert liver specialists. As this is a specialist hepatology centre, staffed by nationally recognised hepatologists, we did not impose a minimum number of tests for a diagnosis of advanced fibrosis, but rather left the judgement to the individual specialist. In previous evaluations of decision-making, we have found that these judgements are maybe based on a minimum set or numerous test results.

Each patient was assessed by an expert hepatologist, with access to several fibrosis tests including liver biopsy and FibroScan. We therefore believe that it is justified to base the diagnosis of advanced fibrosis on the expert hepatologist's documented decision in clinic.

In response to the reviewer's comments, we have taken the opportunity to follow his advice by further detailing the modes of diagnosis, and so have now further clarified how many patients were diagnosed based on biopsy, imaging and FibroScan and hope this improves the manuscript:

Several patients had more than one test. As described above, decisions on fibrosis diagnosis are often based on a composite of several tests – it is therefore not possible to choose a single test for each patient that yielded the diagnosis.

Of the patients referred with suspected ArLD who had advanced fibrosis (82/229), the frequency with which these tests were used was: liver biopsy in 10% (8/82); FibroScan in 41% (34/82); and radiology in 62% (51/82). Of the patients referred with suspected NAFLD who had advanced fibrosis (88/529), the frequency with which these tests were used was: liver biopsy in 47% (41/88); FibroScan in 64% (56/88) and radiology in 33% (29/88).

We have added the above two short paragraphs to the results section under a new subtitle 'prevalence of advanced fibrosis in patients referred with suspected ArLD or NAFLD', page 12 lines 12-22. In doing so, we have moved the first paragraph from the next section (entitled 'risk of advanced fibrosis in patients referred with suspected ArLD) up to the new section where it is now the first paragraph. This means these two results sections now better reflect their subtitles.

10) *Choice of TE cut-off: always some controversy around this, however very important in particular for cross study comparison. The recent multi etiology large scale validation study (PMID: 33307138) provides some clarity and support 12Kpa as a general cut-off to rule-in advanced cirrhosis and 7kpa to rule out. With the current development in the NASH field and health care pathways some places towards treatment of lifestyle factors, significant fibrosis (\geq F2) is also relevant and should be included in the main analysis – i.e. cut-off of 8kpa.*

We agree that there are uncertainties around optimal FibroScan thresholds, and in particular it has been shown that these should be altered according to the underlying aetiology, and thus we specified a different cut off of 11kpa for ArLD and 10kpa for NAFLD. However, FibroScan staging was not the primary outcome measure of our study. Our primary outcome (diagnosis of advanced fibrosis) was based on a documented decision made by an expert hepatologist using composite clinical judgement based on the integration of a range of methods, as described above and not based on a particular FibroScan threshold. In addition, in the small minority of cases where the diagnosis was ambiguous the assessment of fibrosis severity was made by the first two authors using composite clinical judgement employing the predefined literature-based thresholds for FibroScan. We donot think it would be correct to revise our analysis post-hoc based on a new paper-in-press. Furthermore, we do not think that altering the thresholds would materially affect the message of conveyed in our study.

Thank you for highlighting the important paper by Papatheodoridi et al (PMID: 33307138) (1) but we do not think the data presented by these authors pertains to our study for the following reasons. This large-scale study aims to clarify FibroScan thresholds for mixed-aetiology liver disease, and presents data suggesting an improvement in the classification rate for advanced fibrosis from 73% using the

original Baveno VI thresholds to 84%. However only 17% of the study sample had alcohol-related liver disease. Secondly while Papatheodoridi et al. found the <8kpa threshold was effective to 'rule-out' advanced fibrosis in a mixed aetiology cohort, our study focused on patients referred to a specialist liver centre with 'suspected alcohol-related liver disease' where the liver specialists were aiming to 'rule-in' advanced fibrosis in patients with a high pre-test probability of disease. Papatheodoridi et al. suggested a 'rule-in' cut off of 12kpa performed less well in ArLD or NAFLD than in the viral hepatitis, which made up the majority of the cohort. Finally, the authors of this paper note a limitation in that the patients in the alcohol cohort were not necessarily abstinent for at least a week prior to undergoing transient elastography assessment, which may have influenced the results as recent alcohol intake and withdrawal have been shown to significantly affect FibroScan results. The study highlights the need for further prospective validation of thresholds, and we agree that this is a necessity particularly for alcohol-related liver disease, to identify an optimum threshold.

11) Further it must be described what defines an unnecessary referral - <10kpa?

The term 'unnecessary referral' was derived from work done with the local Liver Working Group on referral pathways. The Group included local GPs, Public Health specialists, Healthcare Commissioners, patients and liver specialists. They set out to determine the appropriate location for the management of patients with advanced and less advanced liver disease (2). This working group identified the benefits of referring patients with advanced fibrosis to liver specialists in secondary care, but determined that patients with less advanced fibrosis should remain in primary care where they could be appropriately managed by their GP. National guidelines also now state that screening for advanced fibrosis should take place in primary care, and patients without advanced fibrosis do not need referral to secondary care (3). Hence the group decided that it is unnecessary to refer patients who do not have advanced fibrosis to a liver specialist.

We agree that we could have defined this better in the manuscript, and have now included the sentence below in the methods section under the subtitle 'data collection', page 8 lines 23-25:

'Unnecessary referrals' were defined as those patients that, subsequent to an assessment by a liver specialist, were deemed not to have advanced fibrosis and could be discharged back to ongoing care in the community.

12) page 14/25, line 17: are the limits for alcohol consumption equal for men and women, with 14 units per week?

Yes, the 14-unit limit is the same for men and women, as per the UK national recommendations on safe drinking levels.

13) page 15/25, line 37: Why was blood tests from secondary care used? Was it not available in primary care at the referral point?

Blood tests in primary care were not always available in full in the referral letter, and were often less extensive than those undertaken in hepatology secondary care (for example, some did not have AST or GGT). To reliably have the same snapshot of all available results for all the patients, the first-hepatology-clinic blood results were used.

14) Table 1: provide Number with liver biopsy, number with successful fibroscan and value of FS

Thank you, we have added these numbers to table 1

15) Table 2, also provide data for Fib4 of > 2,67

In our table 2 we used two literature-based pre-defined FIB4 thresholds relevant specifically to ArLD (4, 5). Whilst a threshold of 2.67 has not been validated for ArLD, we are happy to look at this, and enclose the results below. If the editorial team feel it would add to the paper by including this 3rd threshold, we are happy to include it in the manuscript too.

Table 2: Accuracy of indirect fibrosis markers in detecting advanced fibrosis in a cohort of 231 patients referred from primary care with suspected ArLD. (N= 225/231.)

Indirect fibrosis test (n=225/231)	Correctly classifies	Sensitivity (95% CI)	Specificity (95% CI)	PPV (95% CI)	NPV (95% CI)	LR+ (95% CI)	LR- (95% CI)	TP FP	FN TN	False negative rate (%)	False Po rate (%)
APRI ≥ 1	165 (73.3%)	64.6% (54-75)	78.3% (70-85)	63.1% (52-73)	80% (72-86)	3.02 (2.13-4.28)	0.44 (0.33-0.6)	53 31	29 112	35.4	21.7
FIB4 ≥ 3.25	170 (75.6%)	52.4% (41.2-63.5)	88.8% (82.2-93.3)	72.9% (59.5-83.3)	76.5% (69.1-82.6)	4.69 (2.83-7.77)	0.54 (0.43-0.67)	43 16	39 127	47.6	11.2
FIB4 ≥ 2.67	166 (73.97%)	54.9% (43-66)	84.6% (78-90)	66.6% (56-75)	77% (68-80)	3.6 (2.32-5.5)	0.53 (0.42-0.68)	45 22	37 121	45.1	15.4
FIB4 ≥ 1.45	149 (66%)	78% (67.3-86.1)	59.4% (50.9-67.4)	52.4% (43.3-61.5)	82.5% (73.5-89)	1.92 (1.53-2.42)	0.37 (0.24-0.56)	64 58	18 85	22	40.6

Reviewer: 2

Dr. Joern Schattenberg, Johannes Gutenberg University Hospital Mainz

Comments to the Author:

1) This is an important analysis. It needs to be clarified, how alcohol consumption was captured and analyzed in this retrospective chart review. How was the classification for cases with undocumented alcohol consumption or uncertainty about alcohol consumption handled? Which categories did you use to classify in the absence of the documentation of gram / day in the medical records?

Data on alcohol consumption were available for 738/762 participants in the cohort of patients with NAFLD/AFLD. These data were extracted from either the GP referral letter or from the first hepatology clinic letter recorded in the electronic patient records. All data were recorded as number of units per week. Where average daily consumption was recorded quantities were multiplied by 7 to determine the weekly amount. In the 24 cases where accurate information about the amount of alcohol consumed was not available no data on alcohol consumption were included in the analyses. We have made it clear in tables 1 and 2 where any missing data occurs by displaying the denominator for each variable, and only available data were included in analyses throughout.

2) The term BAFLD is not well defined. Has it been adopted by an organization? Is there evidence, that the outcome is different when comparing to NAFLD or ALD? Please provide some rationale for addressing this. How is counselling different - with regards to prevention and if not, why single this out vs for example stratifying by the amount of fibrosis, e.g. non-significant fibrosis vs significant fibrosis? Here treatment and outcomes would be different.

In our study we describe a doubling of the odds of advanced fibrosis in those patients with fatty liver disease who drink 14 units/week or more. These individuals do not meet the diagnostic criteria for NAFLD due to the amount of alcohol they consume, but they have risk factors for fatty liver disease and drink more than recommended amounts of alcohol. Thus, they have risk factors for Both Alcohol and Fatty Liver Disease, hence BAFLD. BAFLD was a term first coined by Glynn-Owen et al. (6) to describe the increasingly recognised patient population where combined liver risk factors of alcohol and obesity/metabolic syndrome overlap, leading to an increased risk of liver fibrosis. A recent systematic review with meta-analysis led by the same study team in Dec 2020 (7) concludes that “compared to normal weight participants drinking alcohol within UK recommended limits, the relative risk of chronic liver disease in overweight participants drinking above limits was 3.32 (95% CI 2.88 to 3.83) and relative risk in obese participants drinking above limits was 5.39 (95% CI 4.62 to 6.29)”(7). This meta-analysis included results of a study by Tremblay et al which included over 95,000 participants (8). We believe that there is considerable benefit in raising the profile of the adverse interaction of alcohol and fat and think that the term BAFLD captures this concept well. While we recognise that the terminology is not yet widely accepted, we think that the term BAFLD will help raise awareness of the interaction of these risks and hopefully improve knowledge mobilization within public health, policy makers and the general public.

3) For discussion.

This reviewer struggles with the term unnecessary referral, as it neglects the need and function to provide counseling in the absence of advanced fibrosis and in patients that the primary needs some "additional" feedback on. The term "misdirected" would catch this better? Meaning another specialist would be needed? If Hepatology declines this referrals, it should be clear where they should be sent to inform referring physicians. While the context (NICE system) is important, a management proposition would be of interest to non-NICE based physicians. Which specialist should deal with these patients? When should they be referred to Hepatology? The reduction on prioritisation of referrals falls short to address the management need that arises. Please put this into perspective. Can you provide the reason for referral? Along this line, did you see all referred patients or is there another filter that would decide to reject a referral based on a decision before reaching the clinic? How many referrals were rejected?

All consecutive referrals with suspected AFLD or NAFLD during the 3-year study period were retrospectively included.

'Suspected ArLD' referrals were defined as those in which the GP referral letter requested an assessment by a liver specialist specifying concerns about suspected ArLD or expressing concerns about a patient's alcohol intake.

'Suspected NAFLD' referrals were defined as those in which the GP referral letter either specified that they were referring the patient to hepatology with 'with suspected NAFLD' or 'on the local NAFLD referral pathway', OR, in the absence of any other cause of liver dysfunction, where the GP specified that the patient had steatosis or chronic liver disease on ultrasound in combination with mentioning metabolic risk factors (BMI \geq 25, diabetes, high waist circumference, high cholesterol or hypertension).

We have now clarified this by adding the above paragraph in the manuscript in the methods section, page 7 (lines 29-33) and page 8 (lines 1-4).

We recognise that the term 'unnecessary referrals' requires explanation and have now included the following definition to clarify in the methods section, page 8, lines 23-25 :

“'Unnecessary referrals' were defined as those patients that, subsequent to an assessment by a liver specialist, were deemed not to have advanced fibrosis and could be discharged back to ongoing care in the community.” This is in accordance with the local pathway of care agreed between primary care physicians, liver specialists with input from public health, commissioners and expert patients, underpinned by national guidelines (3, 8).

NICE and BSG guidelines state that patients need referral to secondary care if they have evidence of advanced fibrosis or cirrhosis, but that screening for these can safely take place in the community with non-invasive tests (3, 9).

Whilst we understand your query about patients deemed 'unnecessary' potentially missing out on seeing a specialist and getting alcohol counselling, we would like to make the following points. Firstly, it has not been shown that patients benefit from getting this counselling from a hepatologist. In fact, previous research has identified nursing staff as best placed to deliver alcohol brief interventions (10), and a study by Eggleston et al found that patients regard advice delivered by nurses as equal to that delivered by doctors (11). Secondly, in having lots of 'unnecessary referrals' to secondary-care (as per NICE and BSG guidelines stated above, patients without advanced fibrosis can remain in primary care), it is not only an inefficient use of resources, but is likely to be putting patients through the stress of unnecessary investigation through the hospital system when they could be safely managed in primary care.

4) Few typos need to be corrected.

Apologies for this and thank you – we have now gone through the manuscript carefully and corrected any typos.

Reviewer: 1

Competing interests of Reviewer: none

Reviewer: 2

Competing interests of Reviewer: None

With my best wishes

William Rosenberg

Prof William Rosenberg
Peter Scheuer Chair in Liver Diseases

REFERENCES

1. Papatheodoridi M, Hiriart JB, Lupsor-Platon M, Bronte F, Boursier J, Elshaarawy O, et al. Refining the Baveno VI elastography criteria for the definition of compensated advanced chronic liver disease. *J Hepatol.* 2020.
2. Srivastava A, GaDirect targeting of risk factors significantly increases the detection of liver cirrhosis in primary care. *er R, Tanwar S, Trembling P, Parkes J, Rodger A, et al. Prospective evaluation of a primary care referral pathway for patients with non-alcoholic fatty liver disease. J Hepatol.* 2019.
3. Newsome PN, Cramb R, Davison SM, Dillon JF, Foulerton M, Godfrey EM, et al. Guidelines on the management of abnormal liver blood tests. *Gut.* 2018;67(1):6-19.
4. Thiele M, Madsen BS, Hansen JF, Detlefsen S, Antonsen S, Krag A. Accuracy of the Enhanced Liver Fibrosis Test vs FibroTest, Elastography, and Indirect Markers in Detection of Advanced Fibrosis in Patients With Alcoholic Liver Disease. *Gastroenterology.* 2018;154(5):1369-79.
5. Moreno C, Mueller S, Szabo G. Non-invasive diagnosis and biomarkers in alcohol-related liver disease. *J Hepatol.* 2019;70(2):273-83.
6. Glyn-Owen K, Bohning D, Parkes J, Roderick P, Buchanan R. The combined effect of alcohol and obesity on risk of liver disease: a systematic review and meta-analysis. *Hepatology.* 2019;70(S1):753A-A.
7. Glyn-Owen K, Bohning D, Parkes J, Roderick P, Buchanan R. The combined effect of alcohol and body mass index on risk of chronic liver disease: A systematic review and meta-analysis of cohort studies. *Liver Int.* 2020.
8. Trembling PM, Apostolidou S, Gentry-Maharaj A, Parkes J, Ryan A, Tanwar S, et al. Risk of chronic liver disease in post-menopausal women due to body mass index, alcohol and their interaction: a prospective nested cohort study within the United Kingdom Collaborative Trial of Ovarian Cancer Screening (UKCTOCS). *BMC Public Health.* 2017;17(1):603.
9. NICE-Guidance. Cirrhosis in Over 16s: Assessment and Management. National Institute for Health and Care Excellence: Guidance. 2016 [Available from: <https://www.nice.org.uk>].
10. Lock CA, Kaner EF. Implementation of brief alcohol interventions by nurses in primary care: do non-clinical factors influence practice? *Fam Pract.* 2004;21(3):270-5.
11. Eggleston J, Gallagher J, Gallagher M, al. e. Who should give lifestyle advice in general practice and what factors influence attendance at health promotion clinics? Survey of patients' views. *Br J Gen Pract.* 1995;45(401):669-71.

VERSION 2 – REVIEW

REVIEWER	Krag, Aleksander Department of Gastroenterology, Odense University Hospital, University of Southern Denmark
REVIEW RETURNED	07-Mar-2021
GENERAL COMMENTS	The authors have convincingly revised the manuscript and overall improved clarity and ease of reading. The

	manuscript has a well-structured flow and the message of the study is much clearer. Thank you for an important study. A few minor issues  1. - 2. - 3. Regarding inclusion and exclusion criteria. Comment: I wonder how you have handled referrals with vague symptoms? Weight loss? Fatigue? abdominal pain? Were these patients included in the study, or did you require more solid indication for liver disease at referral? 4. - 5. Inclusion and exclusion criteria again – It is very good that you have specified the definitions, but does that equal inclusion? I would suggest to add a section/appendix stating the inclusion and exclusion criteria. It is not easy to see it through. But very important for later study comparisons and discussions in other countries to change clinical practice for referral etc 6. - 7. - 8. - 9. Very nice section, a clear improvement to the study, and interesting to me as a reader. I am completely aligned with the fact that the hepatologist can be accounted for correct diagnosis, however, it is interesting as a reader, to see how many have the diagnosis based on a liver biopsy, FibroScan, radiology etc. So thank you very much for this added value to the study. 10. - point taken, the debate of fibroscan cut off is interesting, but not the purpose of this study. 11. Clear improvement – but did you also judge significant fibrosis as unnecessary? 12. - 13. - 14. -
--	--

REVIEWER	Schattenberg, Joern Johannes Gutenberg University Hospital Mainz
REVIEW RETURNED	23-Feb-2021

GENERAL COMMENTS	Congratulations on an important study.
--

VERSION 2 – AUTHOR RESPONSE

Reviewer: 1
Dr. Aleksander Krag, Department of Gastroenterology, Odense University Hospital

Comments to the Author:
The authors have convincingly revised the manuscript and overall improved clarity and ease of reading. The manuscript has a well-structured flow and the message of the study is much clearer. Thank you for an important study. A few minor issues

1. -

2. -

3. Regarding inclusion and exclusion criteria. Comment: I wonder how you have handled referrals with vague symptoms? Weight loss? Fatigue? abdominal pain? Were these patients included in the study, or did you require more solid indication for liver disease at referral?

Patients were only included if their primary reason for referral satisfied inclusion and exclusion criteria for either suspected ArLD or NAFLD. Thus, referrals for general symptoms such as those listed by the reviewer alone without the history meeting the inclusion criteria for ArLD or NAFLD would not have justified inclusion. In response to previous peer review, we amended the definitions for suspected ArLD and NAFLD inclusion criteria (See page 7 lines 29-33):

‘Suspected ArLD’ referrals were defined as those in which the GP referral letter requested an assessment by a liver specialist, specifying concerns about suspected ArLD, or expressing concerns about a patient’s alcohol intake.

‘Suspected NAFLD’ referrals were defined as those in which the GP referral letter either specified that they were referring the patient to hepatology with ‘with suspected NAFLD’ or ‘on the local NAFLD referral pathway’, OR, in the absence of any other cause of liver dysfunction, where the GP specified that the patient had steatosis or chronic liver disease on ultrasound in combination with mentioning metabolic risk factors (BMI \geq 25, diabetes, high waist circumference, high cholesterol or hypertension).

Patients were excluded if they had a pre-existing liver diagnosis of any aetiology, including liver cancer, prior to referral to the hepatology clinic, or if they were already under the care of a Hepatologist or Gastroenterologist for investigation or management of a liver condition.

For clarity, we have now also included a supplementary table reporting the inclusion/exclusion criteria. (Supplementary table 2).

4. -

5. Inclusion and exclusion criteria again – It is very good that you have specified the definitions, but does that equal inclusion? I would suggest to add a section/appendix stating the inclusion and exclusion criteria. It is not easy to see it through. But very important for later study comparisons and discussions in other countries to change clinical practice for referral etc

Many thanks for this, we have taken up your suggestion of adding in a supplementary table to clarify the inclusion and exclusion criteria. This can now be found as ‘supplementary table 2’. We hope this now makes it clearer to read.

6. -

7. -

8. -

9. Very nice section, a clear improvement to the study, and interesting to me as a reader. I am completely aligned with the fact that the hepatologist can be accounted for correct diagnosis, however, it is interesting as a reader, to see how many have the diagnosis based on a liver biopsy, FibroScan, radiology etc. So thank you very much for this added value to the study.

10. - point taken, the debate of fibroscan cut off is interesting, but not the purpose of this study.

11. Clear improvement – but did you also judge significant fibrosis as unnecessary?

Thank you. We followed the BSG and NICE guidelines by considering greater than F2 (advanced fibrosis) as worthy of further investigation and management by the specialist. The guidelines state that patients without evidence of advanced fibrosis (including significant fibrosis/METAVIR F2-equivalent) do not require referral to hepatology specialists, and should instead have further follow-up reviews at suggested defined periods for re-assessment for fibrosis (1, 2).

12. -

13. -

14. -

Reviewer: 2
Dr. Joern Schattenberg, Johannes Gutenberg University Hospital Mainz

Comments to the Author:
Congratulations on an important study

VERSION 3 – REVIEW

REVIEWER	Krag, Aleksander Department of Gastroenterology, Odense University Hospital, University of Southern Denmark
REVIEW RETURNED	30-Mar-2021
GENERAL COMMENTS	I have not further- it has become a very nice an important manuscript.